# Calculating Study on Properties of Al (111)/6H-SiC (0001) Interfaces

**Changqing Wang [1,2]** **, Weiguang Chen [3], Yu Jia [4] and Jingpei Xie [1,\***

[1]   Collaborative Innovation Center of Nonferrous Metals of Henan Province, College of Materials Science and Engineering, Henan University of Science and Technology, Luoyang 471023, China; cqw@lit.edu.cn
[2]   Department of Mathematics and Physics, Luoyang Institute of Science and Technology, Luoyang 471023, China
[3]   Quantum Materials Research Center, College of Physics and Electronic Engineering, Zhengzhou Normal University, Zhengzhou 450044, China; chenweiguang@zznu.edu.cn
[4]   Key Laboratory for Special Functional Materials of Ministry of Education, and School of Physics and Electronics, Henan University, Kaifeng 475001, China; jiayu@zzu.edu.cn
\*   Correspondence: xiejp@haust.edu.cn; Tel.: +86-379-6423-1760

**Abstract:** The research elaborates on the mechanical properties at the Al (111)/6H-SiC (0001) interface based on the density functional theory. Because of the difference in atom category at the interface of 6H-SiC (0001), it takes the C-terminated interface and Si-terminated interface into account. As indicated by the gross energy computing results at the two interfaces, the C-terminated Al (111)/6H-SiC (0001) interface demonstrates a greater adhesion force than the Si-terminated counterpart. Throughout detailed analysis on the bonding mechanism, surface hybridization and charge transfer at the Al (111)/6H-SiC (0001) reaction interface, the research reveals its strong covalent characteristics. According to the comparative study on the ideal tensile strength and general stacking fault energy at varying cleavage surfaces, a conclusion can be fitly reached that the fracture at the Al (111)/6H-SiC (0001) interface is easily seen in Al-Al bonds in the Al matrix instead of C(Si)-Al bonds at the interface. Despite the greater adhesion energy of the C-Al bond than the Si-Al bond, Al-Al bonds close to the C-terminated Al (111)/6H-SiC (0001) interface easily fracture due to the low ideal tensile strength.

**Keywords:** first principles; interface; Al matrix composite; tensile; generalized stacking fault energy

## 1. Introduction

Aluminum metal matrix composite (Al-MMC) which is strengthened by silicon carbide (SiC) particles has an edge in multiple aspects, including high strength, high rigidity, high elastic modulus, great durability, favorable anticorrosion properties and thermal stability. For these reasons, Al-MMC has been widely applied in the industries of aircrafts, shipping and automobiles recently [1–5].

For the SiC reinforced aluminum metal matrix composite, its mechanical properties mainly depend on the Al matrix, the microstructure of the SiC reinforcement and the interface structure between the SiC and the Al matrix. As a bridge between the Al matrix and the SiC reinforcement, the interface can not only transfer the load but also adjust the local stress concentration of the composite [6,7]. Additionally, the proper interface state can effectively prevent the fracture of the composite crack growth in the process of cracking. Therefore, it is of great academic significance to investigate the interface structure between the SiC particles and the Al matrix as well as reveal its mechanism for the performance improvement and the design of the aluminum matrix composite.

By employing the quantum chemical method, adhesion behaviors of the $\alpha-$SiC (0001)/Al interfaces have been studied [8]. As indicated by the research results, the bond strength of the SiC-Al combination

is much stronger than that of the Al-Al combination. In recent years, substantial experimental [9] and theoretical [10,11] studies have been performed to analyze the Schottky barrier height of metals at the 6H-SiC (0001) interface. The first-principles method is then applied to observe structural stability and electronic performance, as well as the influence of additions to interfacial adhesive strength at the Al (111)/4H-SiC (0001) interface [12]. Existing findings verify the conductive role of introduced Ti and Si in reinforcing the adhesion strength at the C-terminated Al (111)/4H-SiC (0001) interface as well as the adverse role of Mg and Cu. Previously, our group conducted a theoretical study on the point defect and structural effects at the Al (111)/4H-SiC (0001)interface [13,14]. In this work, the mechanical properties, especially stacking fault and shear behaviors at the Al (111)/6H-SiC (0001) interface, were investigated. In addition, we studied the mechanical properties of the Al (100)/6H-SiC (0001) interface [15]. However, Al (111)/6H-SiC (0001) reaction interfaces have been observed experimentally [16]. In order to study the bonding characteristics of the Al/SiC interface more systematically, the mechanical properties, especially stacking fault and shear behaviors at the Al (111)/6H-SiC (0001) interface, were researched in this paper.

With the objective of better rendering the structural transformation in the aluminum metal matrix composite at the Al/SiC interface, this paper used the first principles method to check the stretching, stacking fault and shear properties at the Al (111)/6H-SiC (0001) interface. Exploration of the ideal tensile strength and the generalized stacking fault (GSF) energy close to the atomic layer at the Al (111)/6H-SiC (0001) interface revealed the fracture vulnerability of Al-Al bonds at the Al/SiC interface instead of C (Si)-Al bonds. In-depth study of the Al/SiC interface combination enables the understanding of how to improve the dynamic performance of the aluminum metal matrix composite material.

## 2. Details of Calculation Methods

The Vienna ab-initio simulation package (VASP) code was used in the gross energy computing operation in the research [17,18]. The computing of density functional theory (DFT) was performed in accordance with the projector augmented-wave (PAW) [19,20] method and GGA exchange correlation function in the form of Perdew–Burke–Ernzerhof (PBE) [21]. For converging related quantities, overall computing operations involved the developed electronic wave function on a plane wave with an energy cutoff value of 600 eV. Sampling was performed by $15 \times 15 \times 1$ Monkhorst–Pack (MP) grid [22] K-points in the irreducible Brillouin zone. Subsequently, the electronic convergence tolerance between every two electronic relaxation steps was preset as $10^{-5}$ eV. As for structural relaxation, the force tolerance of each atom was $10^{-2}$ eV/Å.

The computing of Al and 6H-SiC bulk properties preceded the survey of interface behaviors. Consistent with available findings, the research set the lattice constant of bulk Al as 4.044 Å [23]. Likewise, the computed lattice constant a = b = 3.095 Å and c = 15.177 Å for 6H-SiC complied with the experimental value [24]. The computing results verified the high accuracy of the used parameters. The 6H-SiC (0001) belongs to a classical polar surface which only contains one atom at the surface layer, i.e., the C- or Si- terminated interface. Besides that, there are diverse bonding modes among surface atoms with one or three dangling bonds. A greater stability of the one-dangling-bond surface compared with the three-dangling-bond surface was also verified by the research. As a result, the following sections prioritize the one-dangling-bond surface slab.

The recent work presents that the surface energy of the Al (111) slab with 7 atomic layers can converge to a certain value [12]. Therefore, the supercell used in the present study contained nine Al atomic layers and six SiC layers (as shown in Figure 1). The coherent interface model was chosen here to cater to the periodic boundary conditions in supercell computing. We chose the $1 \times 1$ Al slab along the [−110] and [−101] base vectors to match the $1 \times 1$ SiC slab. In this case, lattice mismatch was about 7.6%. Hence, the softer Al matrix along both the [−110] nd [−101] directions was stretched to cohere with SiC. The whole supercell was completely relaxed in the following calculations. For testing the nonsensitivity of the research results to interface artificial strain energy, the strain between Al and SiC

was redistributed for a comparative study. Under different strain conditions, the variation in the ideal adhesion energy and the generalized stacking fault energy was less than 1%, suggesting that the effect of strain energy on the ideal adhesion energy and the generalized stacking fault energy for the current interfacial structure could be disregarded. In the attempt to eliminate the interplay between adjacent cells, this research placed a supercell with a length of 50 Å in the z direction, and preserved a vacuum area above 15 Å. Passivity of the hanging bonds of atoms was produced by hydrogen atoms at the SiC (0001) surface layer. According to the relative position between the C (Si) atom of the SiC (0001) and the Al atom of the Al (111) at the interface, the research took three stacking sequences "FCC", "HCP" and "TOP" into consideration. Where the C (Si) atom at the interface is above the third Al atom away from the interface, we call the stacking sequence "FCC" (as shown in Figure 1a,d). Where the C (Si) atom at the interface is above the second Al atom away from the interface, we call the stacking sequence "HCP" (as shown in Figure 1b,e). Where the C (Si) atom is on the top of Al atom at the interface, we call the stacking sequence "TOP" (as shown in Figure 1c,f).

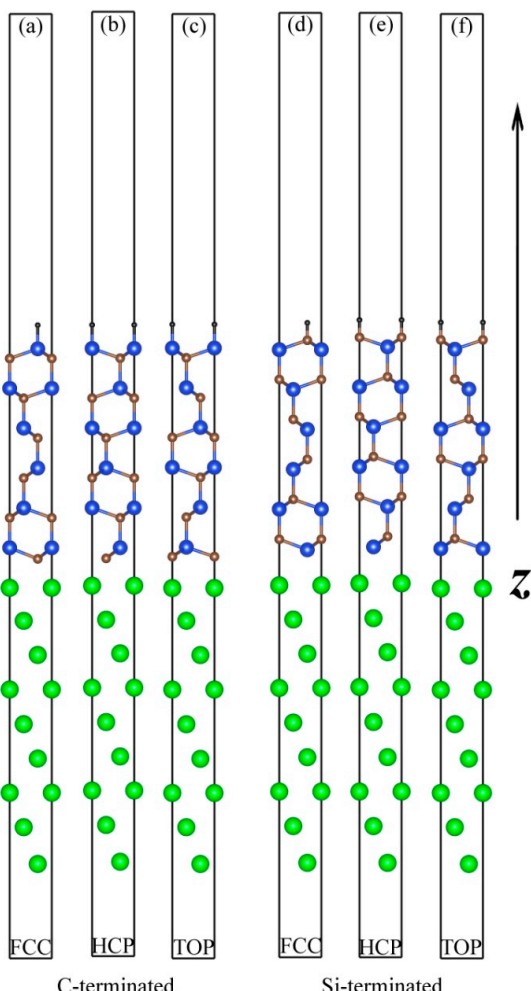

**Figure 1.** Side view of the Al (111)/6H-SiC (0001) interface: C-terminated and Si-terminated interface. There are three stacking sequences at the Al (111)/6H-SiC (0001) interface, namelythe FCC site (**a**,**d**), HCP site (**b**,**e**) and TOP site (**c**,**f**). Where the C (Si) atom at the interface is above the third Al atom away from the interface, we call the stacking sequence "FCC" (as shown in Figure 1a,d). Where the C (Si) atom at the interface is above the second Al atom away from the interface, we call the stacking sequence "HCP" (as shown in Figure 1b,e). Where the C (Si) atom is on the top of Al atom at the interface, we call the stacking sequence "TOP" (as shown in Figure 1c,f). Al, C, Si and H atoms are expressed by green, brown blue and black solid spheres, respectively.

## 3. Results and Discussion

### 3.1. Atomic Structures of the Al (111)/6H-SiC (0001) Interfaces

As shown in Figure 1, six types of structures of the Al (111)/6H-SiC (0001) interface were relaxed based on the Density Functional Theory (DFT) method. The relaxed structural parameters of the interfaces have been listed in Table 1, displaying that the equilibrium interface distance $d_0$ (<2 Å) at the C-terminated interface was shorter than that (>2 Å) at the Si-terminated interface. Additionally, the length of C-Al bonds at the C-terminated interface was also shorter than the length of Si-Al bonds at the Si-terminated interface. When the C atom of the SiC (0001) was on the top of the Al matrix atom, the corresponding C-Al bond length, 1.99 Å, was the shortest one. Thus, it can be seen that the combination at the C-terminated interface is stronger in comparison with that at the Si-terminated interface.

Ideal adhesion energy is an important factor for investigating interfacial properties. It can be defined as

$$E_{ad} = \frac{E_{Al(111)} + E_{SiC(0001)} - E_{total}}{S} \tag{1}$$

where $E_{Al(111)}$, $E_{SiC(0001)}$ and $E_{total}$ denote the energy of the separate Al(111), SiC (0001) and the relaxed interface, respectively. $S$ denotes the area of the interface. From Table 1, it can be shown that the ideal adhesion energy of the "TOP" stacking C-terminated interface is the maximum (3.90 J/m$^2$). In general, the greater the adhesion energy, the stronger the interface bonding is. In other words, when the C atom of the SiC (0001) is on the top of the Al atom, the interfacial adhesion work is maximum. Then, the interface combination is the strongest.

**Table 1.** Equilibrium interface distances $d_0$ (Å), C (Si)-Al bonds $r$ (Å) and the ideal adhesion energies $E_{ad}$ (J/m$^2$) at the Al (111)/SiC (0001) interface.

| Interfaces | | $d_0$ (Å) | $r$ (Å) | $E_{ad}$ (J/m$^2$) |
|---|---|---|---|---|
| | TOP | 1.99 | 1.99 | 3.90 |
| C-terminated | HCP | 1.74 | 2.49 | 2.67 |
| | FCC | 1.83 | 2.56 | 2.27 |
| | TOP | 2.53 | 2.53 | 2.93 |
| Si-terminated | HCP | 2.32 | 2.92 | 2.29 |
| | FCC | 2.25 | 2.88 | 2.15 |

### 3.2. Electronic Structures of the Al (111)/6H-SiC (0001) Interfaces

Figure 2 displays the charge density difference at the Al (111)/6H-SiC (0001) interface so as to survey the bonding conditions of the interface. The following equation definesthe charge density difference as below:

$$\rho_{diff} = \rho_{Al(111)/SiC(0001)} - \rho_{Al(111)} - \rho_{SiC(0001)} \tag{2}$$

where $\rho_{Al(111)/SiC(0001)}$, $\rho_{Al(111)}$ and $\rho_{SiC(0001)}$ represent the charge density of the relaxed Al (111)/SiC (0001)interface, the Al (111) and the SiC (0001), respectively. According to the above definition, red represents charge enrichment and yellow stands for charge deficiency in Figure 2. The more charge transfer to the interface, the stronger interface bonding. Based on Figure 2, there was an apparent charge transfer in the six interface structures at the Al (111)/6H-SiC (0001) interface. Charge density depletion took place in sites around the interfacial C (Si) and Al matrix atoms. As a result, these charges in the interface region were from both side atoms. The C (Si) atom at the interface formed a covalent bond with the Al atom of the matrix. Therefore, it can be inferred that the SiC and Al matrix can be well combined. A clean Al-SiC interface was found experimentally by the High Resolution Transmission Electron Microscope (HRTEM) [16].

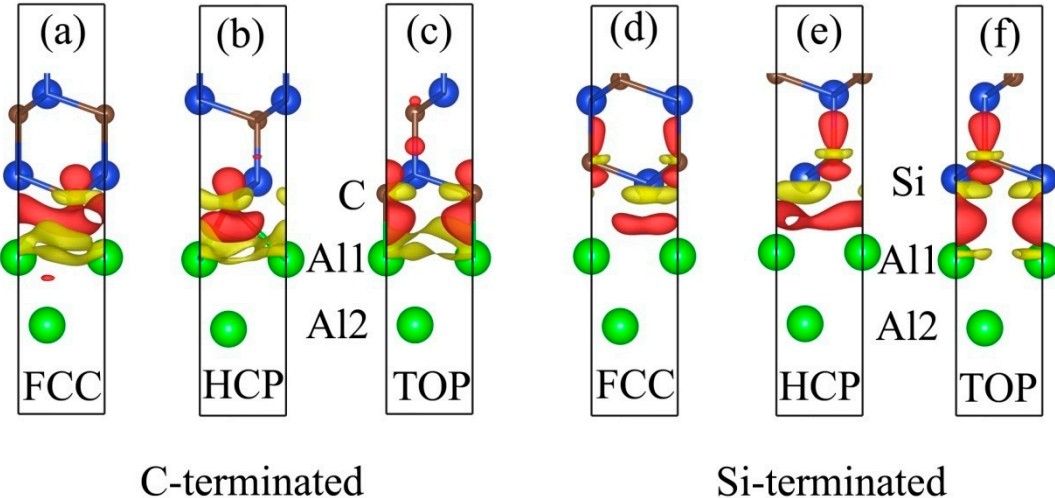

**Figure 2.** Charge density difference of the Al (111)/6H-SiC (0001) interfaces. Red and yellow denote charge enrichment and deficiency, respectively. There are three stacking sequence at the Al (111)/6H-SiC (0001) interface, including the FCC site (**a**,**d**), HCP site (**b**,**e**) and TOP site (**c**,**f**). Al, C, Si and H atoms are expressed by green, brown, blue and black solid spheres, respectively.

To further understand the hybridization of the electronic orbital of the C (Si) atom and the matrix Al atom at the interface, the local electronic density of state (LDOS) was calculated by employing the DFT method. Since the six types of interfaces are similar, here we only calculated LDOS (as shown in Figure 3) of the TOP site of the interface. As presented in Figure 3, the LDOS of the C-terminated interface is shown in the (a), (b) and (c) insert maps. The (g), (h) and (i) insert maps in Figure 3 present the LDOS of the Si-terminated interface. For comparison, the densities of state of the bulk Al (d) and SiC (e,f) are also shown in Figure 3. It can be judged that the electronic density of state of the interface Al atom (Figure 3a,g) near the Fermi level had little change in comparison with that of the bulk Al (Figure 3d). However, for the C (Si) atom at the interface, as presented in Figure 3b,h, the density of state of the p electron near the Fermi level changed greatly compared with that of the bulk SiC (Figure 3e,f), revealing the characteristics of metallization. The result shows that the interface bonding depended mainly on the p-electrons of the interface atoms.

### 3.3. Mechanical Properties of the Al (111)/6H-SiC (0001) Interfaces

The ideal tensile strength or maximum theoretical tensile strength of the material refers to the minimum stress required by the plastic deformation of infinite dislocation free crystals. The key to understanding the limits of the mechanical strength of the nanostructure material, like multilayer film, is to precisely assume the ideal tensile strength. Then, the ideal tensile strength at the Al (111)/SiC (0001) interface was computed. Ideal tensile strength simulations employed a relaxed supercell composed of 9 Al atomic layers and 6 SiC atomic layers with a vacuum.

Two TOP site interfaces (as shown in Figure 1c,f) were taken into consideration, containing the relaxed Al (111)/SiC (0001) interface with C or Si termination at the interface. During the simulation process, the SiC layer moved upward rigidly by 0.5 Å while fastening two Al layers at the bottom part. For any displacement, atoms inside the fixed area can only relax at (111) plane, but comparatively speaking, those inside the free area may relax in all directions. The energy of the supercell increased when the supercell was stretched from its equilibrium position. After concluding the solution of energy displacement curve at each point, the research then divided it by interface area, and thus derived the tensile stress at each displacement level.

The stress–displacement curves of different cleavage plane near the Al (111)/SiC (0001) interface during the tensile strength simulations are shown in Figure 4. At first, tensile strength increased together with displacement and reached the maximum before mechanical fracture, thus gaining ideal

tensile strength of interface. The equilibrium interfacial distance ($d_0$) and the ideal tensile strength ($\sigma$) at different cleavage planes are summarized in Table 2. From Table 2, the ideal tensile strength at the C-Al1 interface was 31.65 GPa, greater than that of other cleavage planes at the C terminated Al (111)/SiC (0001) interface. This is due to the larger adhesion energy of the C-terminated interface. Besides, it is interesting that the ideal tensile strength of the dissociated surface (such as the Al1-Al2, Al2-Al3, Al3-Al4 andAl4-Al5 interfaces) in the Al matrix was less than that of the bulk Al, 14.68 GPa. However, for the Si terminated interface, the ideal tensile strength of the Al1-Al2 cleavage plane was greater than that of the bulk Al. For the Al (111)/SiC (0001) interface, it was found that the fracture easily occurred at the Al-Al bond in the Al matrix instead of the C (Si) -Al bond.

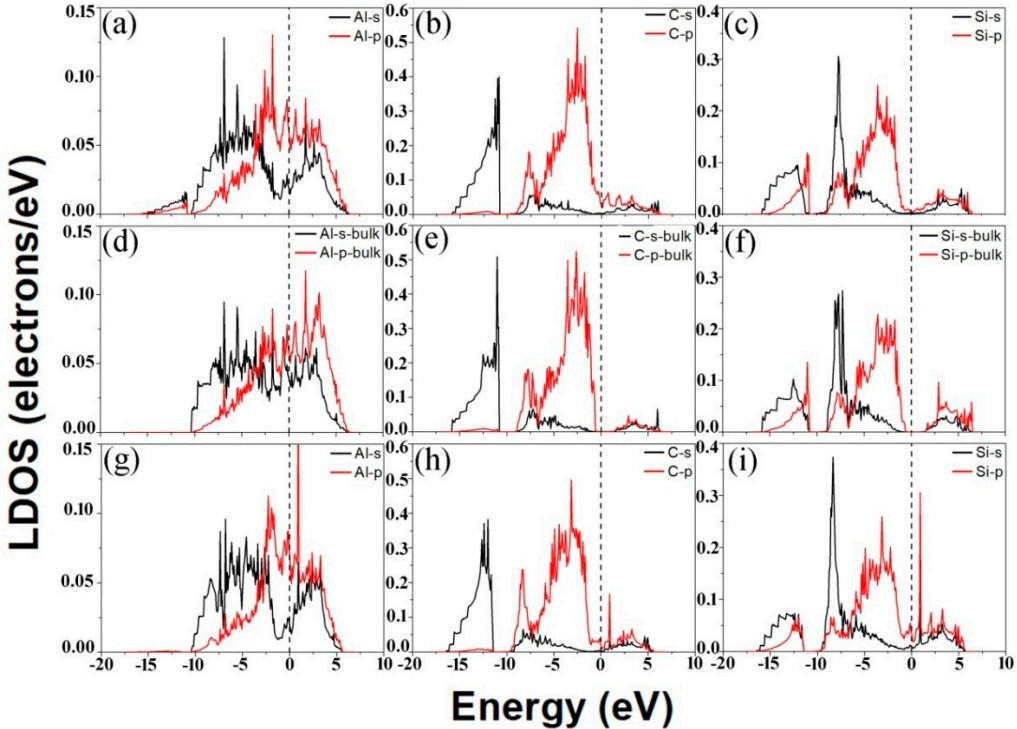

**Figure 3.** Local electronic density of state (LDOS) of the TOP site of the C-terminated (**a**–**c**) and the Si-terminated (**g**–**i**) Al (111)/6H-SiC (0001) interface. For comparative purposes, the LDOS of the bulk of Al (**d**) and SiC (**e**,**f**) are shown. As shown in the dotted line, the Fermi energy shifted to 0 eV. Black and red solid lines represent densities of state of the s and p orbital electrons, respectively.

Based on the concept of generalized stacking fault (GSF) energy, it is feasible to identify the coherent interface structure under stable thermal conditions. In a broad sense, GSF energy is considered as the extra energy per unit area required by rigid shear displacement on a glide plane. Additionally, through differentiating stable and unstable stacking faults [25], GSF energy plays a crucial role in recognizing the possibility of dislocation dissociation reactions. Figure 5 displays the GSF energy curve with displacement at varying Al layers in the <11-2> direction at the Al (111)/SiC (0001) interface. Table 2 shows the computing value of generalized stacking fault energy. Our DFT-based stacking fault energy of the bulk Al, 0.17J/m$^2$ is consistent with the former calculated value (0.142 J/m$^2$ [26]) and experimental value (0.166 J/m$^2$ [27]). According to Table 2, the stacking fault energy of the C-Al interface was 1.32 J/m$^2$, more than twice that of the Si-Al interface, 0.56 J/m$^2$.Starting from the Al1-Al2 plane, the GSF energy curve at Al layer away from the interface was close to that of the bulk Al with intrinsic stacking fault at the Shockley partial displacement<11-2>$a_0$/6).

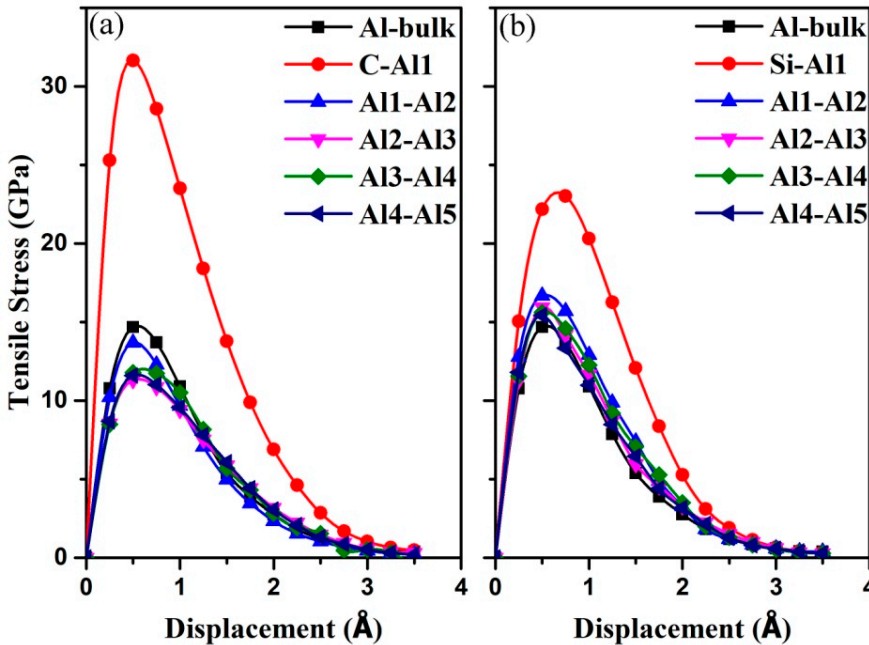

**Figure 4.** The stress–displacement curves of the C-terminated (**a**) and Si-terminated (**b**) Al (111)/6H-SiC (0001) interfaces. C-Al1 represents the cleavage plane between C and Al1 atomic layers, as shown in Figure 2. The same is true of other situations. For comparison, the stress–displacement curve of the bulk Al is also shown here.

**Table 2.** Equilibrium interfacial distance ($d_0$), ideal tensile strength ($\sigma$) and generalized stacking fault energy ($\gamma$) at different cleavage planes. C-Al1 is expressed as the cleavage plane between C and Al1 atomic layers in Figure 1. The same is true of other cases.

| Interfaces | | $d_0$ (Å) | $\sigma$ (GPa) | $\gamma$ (J/m$^2$) |
|---|---|---|---|---|
| C-terminated | C-Al1 | 1.99 | 31.65 | 1.32 |
| | Al1-Al2 | 2.08 | 13.68 | 0.23 |
| | Al2-Al3 | 2.18 | 11.30 | 0.18 |
| | Al3-Al4 | 2.20 | 11.78 | 0.12 |
| | Al4-Al5 | 2.32 | 11.59 | 0.14 |
| Si-terminatedbulk | Si-Al1 | 2.53 | 22.18 | 0.56 |
| | Al1-Al2 | 2.09 | 16.68 | 0.21 |
| | Al2-Al3 | 2.17 | 15.94 | 0.19 |
| | Al3-Al4 | 2.25 | 15.61 | 0.13 |
| | Al4-Al5 | 2.32 | 15.44 | 0.15 |
| | - | 2.33 | 14.68 | 0.17 |

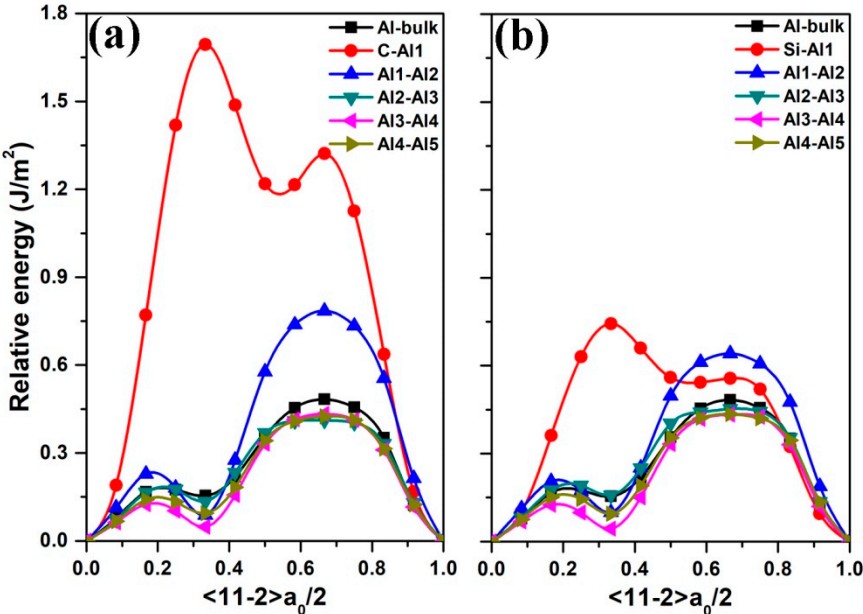

**Figure 5.** The generalized stacking fault (GSF) energy curves for displacements in the (111) plane along the <11-2> direction of different dissociation surfaces of the C-terminated (**a**) and Si-terminated (**b**) Al (111)/6H-SiC (0001) interfaces. C-Al1 represents the cleavage plane between C and Al1 atomic layers, as shown in Figure 2. The same is true of other situations. For comparison, the generalized stacking fault (GSF) energy curves of bulk Al is also shown here. $a_0$ is the lattice constant of Al.

## 4. Conclusions

This paper performed the first principles method to conduct a systematic study on the electronic structure and dynamic performance at the Al (111)/SiC (0001) interface. Research results prove the strong interfacial bonds and adhesion capacity of the C-terminated interface compared with the Si-terminated interface. By comparing the ideal tensile strength and stacking fault energy of different cleavage surfaces, it can be easily seen that fracture often occurs at Al-Al bonds rather than C (Si)-Al bonds. Even if C-Al bonds at the C-terminated interface are stronger than Si-Al bonds at the Si-terminated interface, Al-Al bonds easily fracture at the C-terminated Al (111)/SiC (0001) interface due to its low ideal tensile strength.

**Author Contributions:** Calculation implementation, C.W. and W.C.; Investigation and analysis, J.X. and Y.J. Calculation data processing, C.W. All authors have read and agreed to the published version of the manuscript.

**Funding:** This research was funded by Natural Science Foundation of China (Grant No. U1604251,51802139) and the Key Scientific Research Projects Plan of Henan Higher Education Institutions (Grant No. 17A510012, 18B140008) and The Plan of Young Backbone Teachers in Colleges and Universities of Henan Province (Grant No. 2019GGJS243).

**Conflicts of Interest:** The authors declare no conflict of interest.

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
