# Peer review of "Calculating Study on Properties of Al (111)/6H-SiC (0001) Interfaces"

_metals, doi:10.3390/met10091197_

Round 1

Reviewer 1 Report

In this paper, the authors have investigated the Al(111)/6h-SiC(0001) interfaces using the first-principle density functional theory method. The calculated results are well explained, but there exists some sloppiness in the text (e.g. Table 1 is missing). The paper can be accepted for publication after addressing the following comments:

(i) The authors need to rewrite the introduction part of the paper focusing on the novelty of their work - How does the current study differ from Wu et. al., Surface Science 670 (2018) 1, and the authors’ previously published work in Mater. Res. Express 6 (2019) 126316?

(ii) The authors state that “In this case, lattice mismatch is about 7.6%. Hence, the softer Al matrix has been both stretched to coherent with SiC.” How do the authors perform relaxation during calculations? Have they fixed the SiC subsystem completely or partially or performed a full relaxation? As the Al subsystem will have a substantial effect on the SiC in the vicinity of the interface, it should be clarified.

(iii) What makes the higher tensile strength of Al-Al bonds in case of a Si-terminated interface than that of the C-terminated interface?

(iv) It would be nice if the authors can calculate the Bader charges and analyze them which will give a better comparison between the systems with different interfaces as well as stacking orders.

Minor comments:

  1. The authors write “Figure” in some places and “Fig.” in other places. Consistency should be maintained.
  2. The charge density plots are not very clear. My suggestion is to remove the atoms far from the interface and just to focus near the interface region to produce a better and enlarged figure.
  3. The labels of the DOS figures are not legible.
  4. 3 caption: “Red and black solid lines represent densities of state of the s and p orbital, respectively.” This is not the case in the figures.
  5. Legend “(b)” is missing in Fig. 4.
  6. What is “a0”?
  7. 5 – x-axis needs unit.

Author Response

1\ According to the reviewer's comments, we have revised the third paragraph of the introduction.

2\ The softer Al matrix along the [−110] and [-101] directions have been both stretched to coherent with SiC. The whole supercell has been completely relaxed in the following calculations. For testing the non-sensitivity of research results to interface artificial strain energy, the strain between Al and SiC is redistributed for comparative study. Under different strain conditions, the variation in the ideal adhesion energy and the generalized stacking fault energy is less than 1%, suggesting that the effect of strain energy on the ideal adhesion energy and the generalized stacking fault energy for the current interfacial structure can be disregarded.

3\ Less charge transfer from Al to Si may make the higher tensile strength of Al-Al bonds in case of a Si-terminated interface than that of the C-terminated interface.

4\ It is a very good idea. We will discuss the charge effects on interface adhesion strength under different doping conditions in the following work.

Minor comments:

1\ We have changed all the Fig. to Figure.

2\ We have removed the atoms far from the interface in the charge density plots (Figure 2).

3\ We have enlarged the labels of the DOS (Figure 3).

4\ We have changed “Red and black solid lines represent densities of state of the s and p orbital, respectively.” to “Black and red solid lines represent densities of state of the s and p orbital, respectively.”

5\ We have added legend “(b)” in Figure 4.

6\ a0 is the lattice constant of Al in the Figure 5.

7\ We have set the total length, <11-2>a0/2 to 1. The x-axis in the Figure 5 does not require the unit.

Reviewer 2 Report

The work is of high scientific quality. The paper is clearly written. The title, abstract and conclusion are within the paper aim. The results are carefully discussed. The paper is scientifically sound and not misleading. The work is relevant and the quality of presentation meets the requirements.

I give a recommendation for the publication of this contribution as is.

Author Response

According to the reviewer's comments, no modification is needed.

Reviewer 3 Report

Manuscript seems to need some modifications, as I suggested in attached pdf.

Font size used in Figure 3 is too small to read. It is recommended to enlarge fonts.

In papers published in Metals, the number of references quoted in main text and figure captions are ordinarily written as [1], not being the form of superscript.

The calculation results clearly indicate that the difference in atomic configuration at Al/SiC interface has a great influence on the properties of the interface. However, explanation or discussion on the reasons of the observed dissimilar phenomena does not seem to be enough for readers to understand.

Other comments can be seen in attached pdf.

Author Response

1\ We have revised the number of references quoted in main text and figure captions.

2\ We have enlarged the Figure 2.

3\We have enlarged the labels of the DOS (Figure 3).

4\ We have added Equation numbers.

5\ According to the reviewer's comments, we have revised our manuscript.

6\ Less charge transfer from Al to Si may make the higher tensile strength of Al-Al bonds in case of a Si-terminated interface than that of the C-terminated interface.

Round 2

Reviewer 1 Report

The authors have satisfactorily addressed the comments.